

# Model simulations of chemical effects of sprites in relation with satellite observations

Holger Winkler[1], Takayoshi Yamada[2], Yasuko Kasai[2,3], Uwe Berger[†], and Justus Notholt[1]

[1]Institute of Environmental Physics, University of Bremen, Germany
[2]Terahertz Technology Research Center, National Institute of Information and Communications Technology, Japan
[3]Department of Environmental Chemistry and Engineering, Tokyo Institute of Technology, Japan
[†]Deceased, 4 April 2019 (Leibniz-Institute of Atmospheric Physics, Kühlungsborn, Germany)

**Correspondence:** H. Winkler (hwinkler@iup.physik.uni-bremen.de)

**Abstract.** Recently, measurements by the Superconducting Submillimeter-Wave Limb Emission Sounder (SMILES) satellite instrument have been presented which indicate an increase of mesospheric $HO_2$ above sprite producing thunderstorms. These are the first direct observations of chemical sprite effects, and provide an opportunity to test our understanding of the chemical processes in sprites. In the present paper, results of numerical model simulations are presented. A plasma chemistry model

in combination with a vertical transport module was used to simulate the impact of a streamer discharge in the altitude range 70–80 km, corresponding to one of the observed sprite events. Additionally, a horizontal transport and dispersion model was used to simulate advection and expansion of the sprite volumes. The model simulations predict a production of hydrogen radicals mainly due to reactions of proton hydrates formed after the electrical discharge. The net effect is a conversion of water molecules into $H + OH$. This leads to increasing $HO_2$ concentrations a few hours after the electric breakdown. According to

the model simulations, the $HO_2$ enhancements above sprite producing thunderstorms observed by the SMILES instrument can not solely be attributed to the detected one sprite event for each thunderstorm. The main reason is that the estimated amount of $HO_2$ released by a sprite is much smaller than the observed increase. Furthermore, the advection and dispersion simulations of the observed sprites reveal that in most cases only little overlap of the expanded sprite volumes and the field of view of the SMILES measurements is expected.

## 1   Introduction

Sprites are large scale electrical discharges in the mesosphere occurring above active thunderstorm clouds. Since Franz et al. (1990) reported on the detection of such an event, numerous sprite observations have been made, e.g. Neubert et al. (2008); Chern et al. (2015). Sprites are triggered by the underlying lightning, and their initiation can be explained by conventional air breakdown at mesospheric altitudes caused by lightning-driven electric fields, e.g. Pasko et al. (1995); Hu et al. (2007).

Electrical discharges can cause chemical effects. In particular, lightning is known to be a non-negligible source of nitrogen radicals in the troposphere, e.g. Schumann and Huntrieser (2007); Banerjee et al. (2014). The chemical impact of electrical discharges at higher altitudes is less well investigated. However, it is established that the strong electric fields in sprites drive plasma chemical reactions which can affect the local atmospheric gas composition. Of particular interest from the atmospheric





chemistry point of view is the release of atomic oxygen which can lead to a formation of ozone, as well as the production of
$NO_x$ ($N + NO + NO_2$), and $HO_x$ ($H + OH + HO_2$), which act as ozone antagonists.

Due to the complexity of air plasma reactions, detailed models are required to assess the chemical effects of sprites. Model simulations of the plasma-chemical reactions in sprites have been presented for sprite halos, e.g. Hiraki et al. (2004); Evtushenko et al. (2013); Parra-Rojas et al. (2013); Pérez-Invernón et al. (2018), as well as sprite streamers, e.g. Enell et al. (2008); Gordillo-Vázquez (2008); Hiraki et al. (2008); Sentman et al. (2008); Winkler and Notholt (2014); Parra-Rojas et al.
(2015); Pérez-Invernón et al. (2020). Almost all of the aforementioned studies focus on short-term effects, and do not consider transport processes. One exception is the model simulation of Hiraki et al. (2008) which accounts for vertical transport and was used to simulate sprite effects on time scales up to a few hours after the electric breakdown event. There is a model study on global chemical sprite effects by Arnone et al. (2014) who used sprite $NO_x$ production estimates from Enell et al. (2008), and injected nitrogen radicals in relation with lightning activity in a global climate-chemistry model. Such an approach is suitable
to investigate sprite effects on the global mean distribution of long-lived $NO_x$. It is not useful for the investigation of the local effects of single sprites in particular regarding shorter-lived species such as $HO_x$.

There have been attempts to find sprite induced enhancements of nitrogen species in the middle atmosphere by correlation analysis of lightning activity with $NO_x$ anomalies (Rodger et al., 2008; Arnone et al., 2008, 2009), but until recently there were no direct measurements of the chemical impact of sprites. A new analysis of measurement data from the SMILES (Super-
conducting Submillimeter-Wave Limb Emission Sounder) satellite instrument indicates an increase of mesospheric $HO_2$ due to sprites (Yamada et al., 2020). These are the first direct observations of chemical sprite effects, and provide an opportunity to test our understanding of the chemical processes in sprites. The model studies of Gordillo-Vázquez (2008); Sentman et al. (2008) predict a sprite induced increase of the OH radical in the upper mesosphere but they do not explicitly report on $HO_2$. The model investigation of Hiraki et al. (2008) predicts a decrease of $HO_2$ at 80 km, and an increase at altitudes 65, 70, and
75 km an hour after a sprite discharge. In the present paper, we show results of sprite chemistry and transport simulations covering a few hours after a sprite event corresponding to the observations of Yamada et al. (2020). The focus of our study lies on hydrogen species, and the model predictions are compared to the observed $HO_2$ enhancements.

## 2   Satellite observations

The SMILES instrument was operated at the Japanese experiment module of the International Space Station. It performed
limb scans up to about 100 km height, and took passive submillimeter measurements of various atmospheric trace gases, e.g. Kikuchi et al. (2010); Kasai et al. (2013). The size of the antenna beam at the tangent point was of the order of 3 km and 6 km in the vertical and horizontal directions, respectively (Eriksson et al., 2014). The analysis of SMILES data by Yamada et al. (2020) shows an increase of mesospheric $HO_2$ over sprite-producing thunderstorms. We provide a brief summary of the results here, further details can be sought from the original article. Three thunderstorm systems have been found for which there
was a sprite observation by the Imager of Sprites and Upper Atmospheric Lightnings (ISUAL, Chern et al. (2003)) on board the FORMOSAT-2 satellite followed by a SMILES measurement in spatial-temporal coincidence with the sprite detection.





Table 1 shows the key parameters of the measurements. In all three cases, the total enhancement of $HO_2$ is of the order of $10^{25}$ molecules inside the field of view of the SMILES instrument. Note that the sprite locations lie outside the SMILES measurement volumes. The shortest horizontal distances between the SMILES lines of sight and the sprite bodies have been

estimated to be about $10\,km$ (events A and C) and $110\,km$ (event B). There is a time lag of 1.5 to 4.4 hours between the sprite detection and the SMILES measurement. Considering typical horizontal wind speeds in the upper mesosphere, Yamada et al. (2020) estimated advection distances of a few $100\,km$ for the sprite air masses during the elapsed times between sprite occurrence and SMILES measurement. Data from the Worldwide Lightning Location Network (WWLLN) indicate strong lightning activity in the respective thunderstorm systems, and Yamada et al. (2020) pointed out that possibly additional sprites

occurred which were not detected by ISUAL.

## 3   Model description

The main tool for our investigation is a one-dimensional atmospheric chemistry and transport model. It is used to simulate the undisturbed atmosphere before the occurrence of a sprite as well as the processes during and after the event. The model's altitude range is 40–120 km, and it's vertical resolution is 1 km. Table 2 shows the modelled species. The model's chemistry

routines are based on a model which has previously been used to simulate short-term chemical effects of sprites (Winkler and Notholt, 2014), and Blue Jet discharges (Winkler and Notholt, 2015). For a proper simulation of the atmospheric chemistry on longer time scales, the reaction scheme of this plasma chemistry model was merged with the one of an atmospheric chemistry model (Winkler et al., 2009) whose reaction rate coefficients were updated according to the latest JPL (Jet Propulsion Laboratory) recommendations (Burkholder et al., 2015). In the following, this model version is referred to as "Model_JPL". For some

reactions also different rate coefficients have been considered. The reason is that there are reports on discrepancies between modelled and observed concentrations of OH, $HO_2$ and $O_3$ in the mesosphere if JPL rate coefficients are used for all reactions (Siskind et al., 2013; Li et al., 2017). In particular, the JPL rate coefficient for the three body reaction

$$H + O_2 + M \rightarrow HO_2 + M, \tag{1}$$

where M denotes $N_2$ or $O_2$, appears to be too small at temperatures of the upper mesosphere. According to Siskind et al.

(2013), a better agreement between model and measurement is achieved if the rate coefficient expression proposed by Wong and Davis (1974) is applied. Therefore, we have set up a model version "Model_WD" which uses the rate coefficient of Wong and Davis (1974) for reaction (1) while all other rate coefficients are as in Model_JPL. Li et al. (2017) have presented different sets of modified rate coefficients for reaction (1) as well as five other reactions of hydrogen and oxygen species. We have tested all these sets of rate coefficients in our model. In the following, we only consider the most promising model version which

uses the 4th set of rate coefficients of Li et al. (2017). This model version is called "Model_Li4". Results of the model version Model_JPL, Model_WD, and Model_Li4 will be compared in Sec. 4.

The effect of the enhanced electric fields occurring in sprite discharges is accounted for by reactions of energetic electrons with air molecules. The electron impact reaction rate coefficients are calculated by means of the Boltzmann solver BOLSIG+





(Hagelaar and Pitchford, 2005), for details see Winkler and Notholt (2014). For the present study, the electron impact reactions

with $H_2O$ and $H_2$ shown in Tab. 3 have been added to the model.

The model has a prescribed background atmosphere of temperature, $N_2$ and $O_2$ altitude profiles. These profiles were derived from measurements of the SABER (Sounding of the Atmosphere using Broadband Emission Radiometry) instrument (Russell et al., 1999). The model uses daily mean day-time and night-time profiles calculated from SABER Level 2A, version 2.0, data for the geo-location of interest. At every sun rise or sun set event, the model's background atmosphere is updated.

The transport routines of the model calculate vertical transport due to molecular and eddy diffusion as well as advection. Details on the transport model can be found in the Appendix A. Transport is simulated for almost all neutral ground state species of the model. Exceptions are $N_2$ and $O_2$ for which the prescribed altitude profiles are used. Transport is not calculated for ions and electronically excited species as their photochemical life-times are generally much smaller than the transport time constants. The abundances of neutral ground state species at the lower model boundary (40 km) are prescribed using mixing ratios of

a standard atmosphere (Brasseur and Solomon, 2005). At the upper model boundary (120 km), atomic oxygen and atomic hydrogen are prescribed using SABER mixing ratios for an altitude of 105 km (The SABER O and H profiles extend only up to this altitude). This causes somewhat unrealistic conditions at the uppermost model levels, see Sec. 4. Furthermore, following Solomon et al. (1982), an influx of thermospheric NO is prescribed at the upper model boundary. For all other species a no-flux boundary condition is applied.

**4   Background atmosphere simulations**

The one-dimensional model was used to simulate the atmosphere prior to sprite event B (Tab. 1). For this purpose, the model was initialised with trace gas concentrations from a standard atmosphere (Brasseur and Solomon, 2005) and then used to simulate a time period of almost ten years before the sprite event. The background atmosphere is made of zonal mean SABER profiles of the latitude stripe $0°–13.5°N$ (symmetric around the sprite latitude of $6.7°N$) for year 2009. For this spin-up run, no

ionisation is included. This allows to use a chemical integration time step as large as one second. Transport is calculated once every minute.

Three model versions with different vertical transport speeds have been tested, see Appendix A for details on the transport parameters. Figure 1 shows modelled mixing ratio profiles in comparison with SABER measurements as well as measurements by the MLS (Microwave Limb Sounder) satellite instrument (Waters et al., 2006). At high altitudes, above, say, 100 km, the

modelled abundances of atomic oxygen and hydrogen are too small compared to SABER profiles. This is a result of using SABER mixing ratios measured at 105 km altitude to prescribe the model's boundary values at 120 km. Test simulations have shown that the sprite altitude region is not significantly affected if different boundary conditions are used, e.g. linearly extrapolated SABER mixing ratios or O and H concentrations from the NRLMSIS-00 model (Picone et al., 2002). The model simulation with fast vertical transport agrees well with the MLS water measurements at altitudes above ∼75 km. On the

other hand, results of this model version differ significantly from the SABER measurements of atomic hydrogen and ozone at altitudes higher than 80 km. The model simulation with medium vertical transport velocities shows a better agreement with





the SABER observations. Therefore, we decided to use this model version for the sprite simulations. The agreement between the model predictions and the measurements is not perfect but reasonable for a one-dimensional model compared to zonally averaged profiles.

The simulation results shown in Fig. 1 were obtained using JPL reaction rate coefficients (Model_JPL). The results of model runs with modified rate coefficients (Model_WD, Model_Li4) do not significantly differ from the results of Model_JPL in terms of the species shown in Fig. 1. However, there are considerable effects of the modified rate coefficients on OH, and $HO_2$. Figure 2 shows altitude profiles of OH and $HO_2$ calculated with Model_JPL, Model_WD, and Model_Li4 in comparison with MLS measurements, and with the SMILES $HO_2$ atmospheric background value for sprite event B. For a comparison of

absolute values, number densities are considered. There are concentration peaks of both OH and $HO_2$ in the altitude range 75–85 km. Location and form of these peaks are affected by the changed reaction rate coefficients, see Fig. 2. The center altitudes of the peaks are highest for the Model_Li4 simulation, and lowest for the Model_WD simulation. While the Model_Li4 agrees well with MLS OH data, it significantly underestimates $HO_2$ compared to the SMILES data point at 77 km altitude. The Model_WD agrees better with the SMILES $HO_2$ measurement, in particular if the vertical resolution of SMILES is taken into

account (Fig. 2). On the other hand, Model_WD predicts too small OH number densities compared to the MLS measurements. It is not possible to draw a conclusion here on what the best set of reaction rate coefficients is. We tend to favor Model_WD as it agrees better with the SMILES $HO_2$ data point than the other model versions. Note that the SMILES measurement corresponds to the actual geolocation and the solar zenith angle of the model profiles whereas the MLS data points are zonal night time averages. Furthermore, the MLS data had to be averaged over a wide latitudinal band (33°S–40°N) to reduce scatter or to

obtain data points at all.

## 5 Sprite chemistry and vertical transport simulations

We have used the model version Model_WD with medium transport velocities for the sprite simulations. The sprite has been modelled as a streamer discharge at altitudes 70–80 km. This is the core region of the considered sprite (see plot (f) in the Supporting Information S1 of Yamada et al. (2020)). Following Gordillo-Vázquez and Luque (2010), a downward propagating

streamer is modelled by an altitude dependent electric field time function which consists of two rectangular pulses, see Fig. 3. The first pulse represents the strong electric fields at the streamer tip, the second pulse represents the weaker fields in the streamer channel (afterglow region). As in the study of Gordillo-Vázquez and Luque (2010), the electric field parameters are based on results of a kinetic streamer model (Luque and Ebert, 2010).

What follows here is a description of the model's parameters in terms of the reduced electric field strength $E/N$, where $E$ is

the electric field strength ($V/cm$) and $N$ is the gas number density ($cm^{-3}$). The reduced electric breakdown field strength is denoted $E_k/N$. Its value is approximately $124\,Vcm^2$. According to Luque and Ebert (2010), the reduced electric field strength at the streamer tip linearly increases with altitude. A value of $3 \times (E_k/N)$ is used at 70 km, and $4.5 \times (E_k/N)$ at 80 km. The model is initialised with an electron density profile from Hu et al. (2007). Charge conservation is accounted for by using the same profile for the initial concentration of $O_2^+$. The streamer tip pulse is switched off when the peak electron density in the





streamer head is reached. Based on the results of Luque and Ebert (2010), the streamer head peak electron density was assumed to scale with air density, and at 75 km altitude a value of $2 \times 10^4$ electrons per $\mathrm{cm}^3$ is used.

Following Gordillo-Vázquez and Luque (2010), the reduced electric field of the second pulse is taken to be $E_k/N$ at all altitudes. Between the two pulses and up to a time of one second after the discharge, $E/N$ has the sub-critical value $30 \times 10^{-17}$ Vcm$^2$. The altitude dependent time-lag between the pulses is determined by the different propagation velocities of

streamer head and streamer tail (Gordillo-Vázquez and Luque, 2010). For the second pulse, a linearly decreasing duration is assumed, ranging from $1.3$ ms at 80 km to $0.3$ ms at 70 km. As can be seen in Fig. 3, with this choice of parameters the modelled conductivities in the streamer channel are close to the value of $3 \times 10^{-7}(\Omega \mathrm{s})^{-1}$ reported in the literature, see Gordillo-Vázquez and Luque (2010), and references therein.

The model was used to simulate a time period of 5.5 hours after the sprite event. It does not appear reasonable to simulate

longer time periods with the one-dimensional model because eventually there is significant horizontal dispersion of the sprite bodies, see Sec. 6. For the first two hours after the breakdown pulse, the full ion and excited species chemistry was simulated. Then, the model switched into the less time-consuming mode without ion-chemistry (like in the spin-up run).

We begin our analysis of the chemical effects by an inspection of charged species. Figure 4 displays the simulated temporal evolution of the most important negative species at 80 km, and at 75 km. At lower altitudes, the general pattern is similar to

the one at 75 km and is therefore not shown. The streamer tip electric field pulse leads to an increase of the electron density mainly due to electron impact ionisation of $\mathrm{N}_2$ and $\mathrm{O}_2$. During the second pulse, $\mathrm{O}^-$ becomes the main negative ion, mainly through the dissociative electron attachment process $\mathrm{e} + \mathrm{O}_2 \rightarrow \mathrm{O}^- + \mathrm{O}$. This is followed by electron detachment reactions, of which the most efficient one is $\mathrm{O}^- + \mathrm{N}_2 \rightarrow \mathrm{e} + \mathrm{N}_2\mathrm{O}$ at almost all altitudes. Only at 80 km, the reaction $\mathrm{O}^- + \mathrm{O} \rightarrow \mathrm{e} + \mathrm{O}_2$ is more important. The process

$$\mathrm{O}^- + \mathrm{H}_2 \rightarrow \mathrm{e} + \mathrm{H}_2\mathrm{O} \qquad (2)$$

does not contribute significantly to the absolute electron detachment rates but plays a role for the hydrogen chemistry (see below). Subsequently, there is a formation of molecular ions, initiated mainly by $\mathrm{e} + \mathrm{O}_3 \rightarrow \mathrm{O}_2^- + \mathrm{O}$ at all altitudes. The resulting relative abundance of molecular ions is small at 80 km but larger at lower altitudes where eventually $\mathrm{CO}_4^-$, $\mathrm{CO}_3^-$, and $\mathrm{Cl}^-$ become the most abundant ions. This is in overall agreement with previous model studies, e.g. Gordillo-Vázquez (2008);

Sentman et al. (2008).

Figure 5 shows the simulated temporal evolution of the most important positive ions at 80 km, and at 75 km altitude. The primary ions resulting from electron impact ionisation of air molecules are $\mathrm{N}_2^+$ and $\mathrm{O}_2^+$. The former undergoes rapid charge exchange (mainly) with $\mathrm{O}_2$, and after about one millisecond $\mathrm{O}_2^+$ has become the principal ion at all altitudes. This stays the same during the second electric field pulse. Eventually, there is a formation of $\mathrm{O}_4^+$ mainly through the three body reaction

$$\mathrm{O}_2^+ + \mathrm{O}_2 + \mathrm{M} \rightarrow \mathrm{O}_4^+ + \mathrm{M}. \qquad (3)$$

The main loss process for $\mathrm{O}_4^+$ are reactions with water molecules:

$$\mathrm{O}_4^+ + \mathrm{H}_2\mathrm{O} \rightarrow \mathrm{O}_2^+(\mathrm{H}_2\mathrm{O}) + \mathrm{O}_2. \qquad (4)$$





What follows is a formation of positive ion cluster molecules. The ion $O_2^+(H_2O)$ undergoes hydration reactions

$$O_2^+(H_2O) + H_2O \rightarrow H^+(H_2O)(OH) + O_2 \tag{5}$$

$$H^+(H_2O)(OH) + H_2O \rightarrow H^+(H_2O)_2 + OH \tag{6}$$

which produces a proton hydrate $H^+(H_2O)_2$, and releases an OH radical. Larger proton hydrates can form via successive hydration:

$$H^+(H_2O)_n + H_2O + M \rightarrow H^+(H_2O)_{n+1} + M. \tag{7}$$

This is a well known mechanism in the D-region of the ionosphere (e.g., Reid (1977)), and was also predicted to take place in
sprite discharges, e.g. Sentman et al. (2008); Evtushenko et al. (2013). According to our model simulations, proton hydrates have become the most abundant positive ions after a few to several seconds (depending on altitude), see Fig. 5. The speed of proton hydrate formation decreases with altitude as the three body reactions (3) and (7) are strongly pressure-dependent, and also because the abundance of water decreases with altitude (Fig. 1).

Recombination of proton hydrates with free electrons release water molecules and atomic hydrogen:

$$H^+(H_2O)_n + e \rightarrow H + n \times H_2O. \tag{8}$$

The same is true for recombination of proton hydrates with atomic or molecular anions. The net effect of the chain of reactions $(3)-(8)$ is:

$$O_2^+ + e + H_2O \rightarrow O_2 + H + OH. \tag{9}$$

As a result, there is a conversion of water molecules into two hydrogen radicals (H + OH).
Next we consider the impact on neutral hydrogen species. Figure 6 shows the decrease of water corresponding to the formation of hydrogen-bearing positive ions and $HO_x$ at 75 km. The effect is similar at other altitudes 70–80 km. The small discontinuities of proton hydrates and $HO_x$ at a model time of two hours is due to the end of the ion chemical simulations (the total hydrogen amount is balanced, though). Figure 6 also displays the results of a model simulation of the undisturbed atmosphere, that is a model run without electric fields applied. Water, hydrogen radicals and several other species change in the no-sprite
model simulation on time scales of hours (this is not a model drift but due to the fact that the night-time mesosphere is not in a perfect chemical steady-state). Therefore, for a proper assessment of the sprite impact, the following analysis focuses on concentration differences between the sprite streamer simulation and the no-sprite simulation. Figure 7 shows the changes of the total amount of hydrogen atoms contained in those hydrogen species which are significantly affected by the sprite discharge at 75 km. During the first few seconds, there is a formation of positive hydrogen-bearing ions, and an increase of $HO_x$ at the
expense of water molecules. After about ten seconds, the increase of $HO_x$ slows down, and water starts to recover.

The processes just analysed take place in the whole altitude range 70–80 km. At the highest altitudes, there is an additional process which affects the hydrogen chemistry. The already mentioned electron detachment process

$$O^- + H_2 \rightarrow e + H_2O \tag{10}$$





causes a conversion of $H_2$ into water molecules. This leads to a decrease of $H_2$ at 80 km compared to the no-sprite simulation,
see Fig. 8. However, the production of $HO_x$ is still mainly due to hydration reactions of positive ions. The formation of $HO_x$
molecules due to reactions of proton hydrates in the streamer at 80 km is smaller than it is at 75 km. The two main reasons for
this are: (1) The total ionisation decreases with altitude (because the streamer tip peak electron density scales with air density);
and (2) the formation efficiency of proton hydrates decrease with altitude (because of pressure dependent three-body reactions,
and decreasing water concentrations). Both aspects can be seen in Fig. 5. Other species than the ones shown in Fig. 8 do
not contribute significantly to hydrogen changes. The reactions of energetic electrons with $H_2$ and $H_2O$ during the discharge
(Tab. 3) are irrelevant.

The temporal evolution of the different $HO_x$ species at 75 km is resolved in Fig. 9. Initially, there is an increase of both OH
and H concentrations due to the ion-chemical decomposition of water molecules while the concentration of $HO_2$ is decreased
compared to the undisturbed atmosphere. The main reason for the latter are reactions of $HO_2$ with increased amounts of atomic
oxygen produced in the sprite streamer:

$$HO_2 + O \rightarrow OH + O_2. \tag{11}$$

On longer time scales, the concentration of $HO_2$ increases. The most important production process at all altitudes is the three
body reaction

$$H + O_2 + M \rightarrow OH + O_2 + M. \tag{12}$$

Figure 10 displays $HO_2$ concentrations in the sprite streamer at altitudes 70, 75, and 80 km. The sprite effect on $HO_2$ at 80 km
is negligible.

In Figure 11 the concentration changes of $HO_2$ and $HO_x$ as a function of altitude are displayed for different times after the
sprite event. Note that after 1.5 hours, the concentration of $HO_2$ is smaller for the sprite model run than for the no-sprite model
run basically at all altitudes 70–80 km. Therefore, according to this model result, the $HO_2$ enhancement observed by SMILES
1.5 hours after sprite event B can not be attributed to that event. A possible explanation could be that other sprites previously
occurred near the SMILES measurement volume. On longer time scales, there is an $HO_2$ enhancement at all altitudes 70–
80 km, and an accumulation of $HO_2$ released by different sprites appears possible. Between 2.5 and 4.5 hours of model time,
the $HO_2$ enhancement is basically constant (Fig. 11). At 77 km (tangent height altitude of the SMILES measurement) the
increase of $HO_2$ is of the order of $10^4$ molecules per $cm^3$. The largest increase of the order $5 \times 10^4$ $cm^{-3}$ is located at altitudes
245 73–74 km.

Up to this point, the model results referred to the concentration changes inside a single sprite streamer. Now we make an
attempt to estimate the resulting total $\Delta HO_2$ inside the SMILES' measurement volume. According to Luque and Ebert (2010),
the streamer diameter at 77 km is ~850 m. A cylindrical streamer volume of 3 km height (vertical extent of the SMILES field
of view) contains about $1.7 \times 10^{19}$ excess molecules $HO_2$. This value for a single streamer section is very small compared
250 to the detected ~$10^{25}$ excess molecules inside the field of view of the SMILES instrument (Tab. 1). Unfortunately, the sprite
images do not allow to infer the number of streamers or a volume filling fraction of the sprite body with streamers. We estimate





these parameters by considering the emissions in the first positive band of molecular nitrogen:

$$N_2(B^3\Pi_g) \rightarrow N_2(A^3\Sigma_u^+) + h\nu. \tag{13}$$

A time integration of the model rates of this process yields a total of $\sim 10^{22}$ photons emitted by one streamer in the altitude
range 70–80 km. Typically, the total number of photons in the first positive band of $N_2$ emitted by a sprite lies in the range
$10^{23}$ to a few $10^{24}$ photons, e.g. Heavner et al. (2000); Kuo et al. (2008); Takahashi et al. (2010). Assuming a value of $10^{24}$
photons emitted by the sprite event under consideration yields $10^{24}/10^{22} = 100$ streamers. This corresponds to a volume filling
fraction of the sprite body with streamers of nearly 10% under the assumption that the sprite was of cylindrical shape with a
diameter of 30 km (Tab. 1). For comparison: Arnone et al. (2014) assumed a higher number of 4500 streamers inside a larger
sprite volume which corresponds to a smaller volume filling fraction of 1%.

The volume of the sprite sampled by SMILES is maximum for a centric intersection of the SMILES antenna beam with the
sprite body. At the tangent point, the antenna beam of SMILES has an elliptical cross section of 3 km in vertical direction,
and 6 km in horizontal direction. For a volume filling fraction of 10%, the measurement volume would contain about $4 \times 10^{20}$
excess molecules $HO_2$. If instead of the $HO_2$ increase at 77 km the five times larger value at altitudes 73–74 km was used,
there would be $2 \times 10^{21}$ excess molecules $HO_2$. Even this value is small compared to the observed $\sim 10^{25}$ molecules.

In conclusion, the estimated modelled enhancement of $HO_2$ due to a sprite is much smaller than the $\Delta HO_2$ observed by
the SMILES instrument. A possible reasons for these discrepancies could be missing chemical processes considered by the
streamer model, inaccurate electric field parameters or reaction rate coefficients. All the results shown here were obtained
with Model_WD. In order to test for the effect of changed rate coefficients, we have also performed sprite simulations using
Model_JPL and Model_Li4. Generally, the results do not differ significantly from the Model_WD simulations. In particular,
the amount of $HO_2$ production is similar in all model versions. However, in case of Model_Li4, the formation of $HO_2$ is faster
than in the other models. As a result, there is already an enhancement of $HO_2$ at a time of 1.5 hours after the electric breakdown
pulse.

Another possible explanation for the differences between our model predictions and observations could be that there was an
accumulation of $HO_2$ produced by a number of sprites. In the thunderstorm systems of interest, several hundreds of lightning
strikes occurred, and it is possible that multiple sprites developed (Yamada et al., 2020). Also, it is known that sprites tend occur
in groups, e.g. McHarg et al. (2002); Hayakawa et al. (2004); Lu et al. (2013). Therefore, an accumulation of $HO_2$ produced by
several sprites appears possible. Note that the measured $HO_2$ enhancements for the three events (Tab. 1) are of the same order
of magnitude although the observed sprites had different sizes, occurred at different distances from the SMILES line of sight,
and there were different time lags between sprite detection and $HO_2$ measurements. Thus, it is unlikely that the measured $HO_2$
enhancements are solely due to the three observed sprites.

However, a large number of sprites would be needed to explain the observed $\Delta HO_2$ provided that the model based estimates
are reliable. For the estimated $4 \times 10^{20}$ ($2 \times 10^{21}$) excess molecules $HO_2$ per sprite event, and assuming an overlap of all
sprite volumes, about 38000 (7600) sprite events would be required. Finally, it has to be emphasized that there are considerable
uncertainties involved in the estimation of the total sprite effects from the model results of a single streamer. For instance,





assuming a number of 4500 streamers per sprite (Arnone et al., 2014) instead of 100 streamers, would significantly increase the sprite's chemical impact.

## 6 Sprite horizontal advection and dispersion simulation

As the SMILES measurements are taken at a few hours after the sprite events, and at distances of several kilometers from
the sprite locations, it is desirable to consider the atmospheric transport processes acting on the sprite air masses. For this purpose, we have applied a Lagrangian plume (or: puff) model. This model calculates the expansion of the sprite body due to atmospheric turbulent diffusion while the sprite center is allowed to move with the wind. Similar approaches were successfully used in research studies on air pollution plumes, aircraft trails, and rocket exhaust, e.g. Egmond and Kesseboom (1983); Denison et al. (1994); Karol et al. (1997); Kelley et al. (2009). Our model accounts only for horizontal transport because
vertical transport is already included in the one-dimensional model run presented in Sec. 5, and more importantly because the time scales of vertical transport in the mesosphere are by orders of magnitude larger than the horizontal ones, e.g. Ebel (1980). The advection of the sprite center is calculated using wind field data originating from the Leibniz-Institute middle atmosphere model (LIMA). LIMA is a global three-dimensional general circulation model of the middle atmosphere (Berger, 2008). It extends from the Earth's surface to the lower thermosphere. In the troposphere and lower stratosphere the model is nudged to
observed meteorological data (ECMWF/ERA-40). LIMA uses a nearly triangular mesh in horizontal direction with a resolution of about 110 km. At each time step, the LIMA wind fields are linearly interpolated to the current position of the sprite center. The expansion of the sprite cross section is calculated by a Gaussian plume model approach, e.g. Karol et al. (1997). The radius of the plume corresponds to the standard deviation $\sqrt{\sigma^2}$ of a Gaussian concentration distribution. If wind shear effects are neglected, the temporal change of the variance $\sigma^2$ is given by (Konopka, 1995):

$$\frac{d\sigma^2}{dt} = 2K \tag{14}$$

with $K$ being the apparent horizontal diffusion coefficient. The formation time of a sprite is short compared to the time scales of atmospheric eddy diffusion. For such an instantaneous source, the diffusion coefficient is given by (Denison et al., 1994):

$$K = K_\infty \left(1 - e^{-t/t_L}\right) \tag{15}$$

where $K_\infty$ is the atmospheric macroscale eddy diffusion coefficient, $t$ is the age of the plume, and $t_L$ is the Lagrangian
turbulence time scale. The latter is connected with $K_\infty$ and the specific turbulent energy dissipation rate $\varepsilon$ thought:

$$t_L = \sqrt{\frac{K_\infty}{\varepsilon}} \tag{16}$$

Based on ranges of literature values of the turbulent parameters for the upper mesosphere (Ebel, 1980; Becker and Schmitz, 2002; Das et al., 2009; Selvaraj et al., 2014) we have considered two cases:

(1) A slow diffusion scenario with $K_\infty = 10^6 \ \mathrm{cm^2 s^{-1}}$, and $\varepsilon = 0.01 \ \mathrm{Wkg^{-1}}$
(2) A fast diffusion scenario with $K_\infty = 2.5 \times 10^7 \ \mathrm{cm^2 s^{-1}}$, and $\varepsilon = 0.1 \ \mathrm{Wkg^{-1}}$




The initial plume diameters were taken to be the horizontal widths of the sprites derived from the sprite observations (Tab. 1). Figure 12 shows results of the plume model simulations for both fast and slow diffusive sprite expansion. Only in case of sprite event C and for the fast expansion scenario, the SMILES field of view lies inside the expanded sprite body. For all other cases there is only little or no overlap of the SMILES measurement volume and the increased sprite volume. This is another

indication that the measured $HO_2$ enhancements cannot (solely) be due to the three observed sprites.

## 7    Summary and conclusions

A plasma chemistry model in combination with a vertical transport module was used to simulate the impact of a single sprite streamer in the altitude range 70–80 km corresponding to an observed sprite event. The model indicates that the most important mechanism for the production of hydrogen radicals are reactions of proton hydrates formed a few to several seconds after the

electrical discharge. The net effect is a conversion of water molecules into $H + OH$. The efficiency of this process decreases with altitude mainly because pressure dependent three-body reactions are involved in the formation process of proton hydrates. At all altitudes, the abundance of $H_2O$ is much larger than the produced amount of $HO_x$. Therefore, water is not a limiting factor for the production of $HO_2$. At all altitudes, the reaction $H + O_2 + M \rightarrow HO_2 + M$ is the most important process for the formation of $HO_2$ after the streamer discharge.

According to the model results, the $HO_2$ enhancements above sprite producing thunderstorms observed by the SMILES instrument can not solely be attributed to the detected one sprite event for each thunderstorm system. The main reasons for this are:

1. The estimated amount of $HO_2$ released by a sprite is much smaller than the observed increase.

2. The time difference of 1.5 hours between the occurrence of the investigated sprite event and the SMILES measurement

of enhanced $HO_2$ concentrations is too short to allow a formation of $HO_2$. (However, a model simulation with modified reaction rate coefficients shows an increase of $HO_2$ already after 1.5 h.)

3. Simulations of the horizontal transport and dispersion of the observed sprites reveal that in most cases only little overlap of the expanded sprite volumes and the field of view of the SMILES measurements is expected.

It is difficult to draw a final conclusion. On the one hand, the model simulations predict an increase of $HO_2$ on time scales

of hours after sprite events supporting the observations. On the other hand, the estimated total numbers are much too small to explain the measured enhancements of $HO_2$. Note, however, that there are considerable uncertainties concerning the calculation of the total sprite effects based on the results of the single streamer simulation. It is not clear whether the discrepancies between model predictions and observations are due to incorrect model parameters and assumptions or whether there are chemical processes missing in the plasma chemistry model. It would be desirable to have more observational data available concerning

the occurrence of sprites and their properties as well as concerning sprite induced chemical perturbations.





## Appendix A: Transport modelling

The transport part of the model calculates the change rate of the number density $n_i$ of species $i$ according to the one-dimensional vertical diffusion and advection equation (Brasseur and Solomon, 2005, e.g.):

$$\frac{\partial n_i}{\partial t} = \frac{\partial}{\partial z}\left[D_i\left(\frac{\partial n_i}{\partial z} + \frac{n_i}{H_i} + \frac{(1+\alpha_T)}{T}\frac{\partial T}{\partial z}\right) + K_{zz}\left(\frac{\partial n_i}{\partial z} + \frac{n_i}{H} + \frac{1}{T}\frac{\partial T}{\partial z}\right)\right] - \frac{\partial}{\partial z}\left(n_i w\right) \tag{A1}$$

with $t$ being time, $z$ altitude, $D_i$ the molecular or atomic diffusion coefficient of species $i$, $K_{zz}$ the vertical eddy diffusion coefficient, $\alpha_T$ the thermal diffusion factor, $T$ the temperature in Kelvin, $H_i$ the individual scale height of species $i$, $H$ the atmospheric scale height, and $w$ the vertical wind speed. The diffusion coefficients $D_i$ (in $\mathrm{cm^2 s^{-1}}$) are given by (Banks and Kockarts, 1973):

$$D_i = 1.52 \times 10^{18}\left[\frac{1}{M_i} + \frac{1}{M}\right]^{1/2}\frac{T^{1/2}}{n} \tag{A2}$$

where $M_i$ and $M$ are the molecular mass of species $i$ and the mean molecular air mass (expressed in atomic mass units), respectively, and $n$ is the air number density (in units of $\mathrm{cm^{-3}}$). Following Smith and Marsh (2005), the thermal diffusion factor is taken to be $\alpha_T = -0.38$ for H and $H_2$, and zero for all other species.

Equation (A1) is solved by an implicit finite difference scheme (Crank and Nicolson, 1996).

The free parameters in Eq. (A1) are the eddy diffusion coefficient $K_{zz}$, and the vertical wind speed $w$. We have experimented
with different altitude profiles of the eddy diffusion coefficient, and decided to use a profile parameterization proposed by Shimazaki (1971):

$$K_{zz}(z) = \begin{cases} A \times \exp\left(-S_1(z-z_0)^2\right) & \text{for } z \geq z_0 \\ (A-B) \times \exp\left(-S_2(z-z_0)^2\right) + B \times \exp\left(S_3(z-z_0)\right) & \text{for } z < z_0, \end{cases} \tag{A3}$$

with standard coefficients $S_1 = S_2 = 0.05 \ \mathrm{km^{-1}}$ and $S_3 = 0.07 \ \mathrm{km^{-1}}$. The parameter $z_0$ is the altitude at which the eddy diffusion is maximal, with $K_{zz}(z_0) = A$. For all model simulations presented here, $A = 10^6 \ \mathrm{cm^2 s^{-1}}$ and $z_0 = 105 \ \mathrm{km}$ were
used. The parameter $B$ controls the eddy diffusion coefficient at lower altitudes. For the three cases "slow", "medium", and "fast" vertical transport (Sec. 4), the following values have been used: $B_{slow} = 3 \times 10^5 \ \mathrm{cm^2 s^{-1}}$, $B_{medium} = 5 \times 10^5 \ \mathrm{cm^2 s^{-1}}$, and $B_{fast} = 1 \times 10^6 \ \mathrm{cm^2 s^{-1}}$.

There are one-dimensional model simulations of the middle atmosphere which do not consider vertical winds but only diffusive transport. We noted that the inclusion of advection due to winds significantly improves the model predictions compared to
satellite measurements. In particular, the abundance of water in the middle to upper mesosphere increases and is in better agreement with observations if upward directed winds are included. This is in accordance with the findings of Sonnemann et al. (2005). Data from the Leibniz-Institute middle atmosphere model (LIMA, see Sec. 6) were used to calculate a vertical wind profile. To reduce scatter, a zonal mean LIMA wind profile for the sprite (event B) latitude 6.7°N of November 2011 was calculated. This LIMA wind profile, however, would cause much too strong transport if it was used in the model in addition
to the diffusive transport. Therefore, the LIMA wind profile was multiplied by a scaling factor $S < 1$ to obtain the profile



for the net vertical wind $w$ in Eq. (A1). A similar approach of scaling wind data originating from a global circulation model to obtain the net vertical wind for a one-dimensional advection-diffusion model was taken by Gardner et al. (2005). For the three cases "slow", "medium", and "fast" vertical transport (Sec. 4), the following values for the scaling factor have been used: $S_{slow} = 0.02$, $S_{medium} = 0.05$, and $S_{fast} = 0.1$.

*Author contributions.* H. Winkler has developed the model, performed the simulations, analysed the results, produced the figures, and is the main author of the article. T. Yamada has provided the SMILES data, and helped with their interpretation. U. Berger has provided the LIMA data, and helped processing them. All co-authors have made significant contributions to the article's text with the exception of Uwe Berger who deceased 4 April 2019.

*Competing interests.* The authors declare that they have no conflict of interest.

*Acknowledgements.* This work was financially supported by the German Research Council (Deutsche Forschungsgemeinschaft – DFG), project number WI 4322/4-1. Parts of the model simulations were performed on the HPC cluster *Aether* at the University of Bremen, financed by DFG in the scope of the Excellence Initiative.





**Table 1.** The three events analysed in Yamada et al. (2020). HW is the horizontal width of the sprites emissions, LT is the local time of the SMILES measurement, $\Delta t$ is the time difference between the sprite observation and the SMILES measurement, $\Delta r$ is the shortest distances between the field of view of the SMILES measurement and the sprite location, TH is the tangent height of the SMILES measurement, and $\Delta HO_2$ is the total enhancement along the line-of-sight of the SMILES measurement,

| Event | Date | Sprite location | HW/km | LT | $\Delta t$/hour | $\Delta r$/km | TH/km | $\Delta HO_2$/molecules |
|---|---|---|---|---|---|---|---|---|
| A | 14 Nov. 2009 | 159.7°W/20.8°N | 17 | 01:15:38 | 2.4 | 10 | 75 | $8.9 \pm 2.5 \times 10^{24}$ |
| B | 18 Nov. 2009 | 78.9°W/6.7°N | 30 | 00:34:06 | 1.5 | 110 | 77 | $16 \pm 2 \times 10^{24}$ |
| C | 9 Mar. 2010 | 19.4°E/1.9°N | 8* | 03:23:52 | 4.4 | 10 | 80 | $17 \pm 2 \times 10^{24}$ |

\*) Only a part of this sprite volume was observed by ISUAL (see Figure 1 in Yamada et al. (2020)).

**Table 2.** Modelled species. The last row shows the molecules additionally included compared to the model of Winkler and Notholt (2015).

| Negative species |
|---|
| e, $O^-$, $O_2^-$, $O_3^-$, $O_4^-$, $NO^-$, $NO_2^-$, $NO_3^-$, $CO_3^-$, $CO_4^-$, $O^-(H_2O)$, $O_2^-(H_2O)$, $O_3^-(H_2O)$, $OH^-$, $HCO_3^-$, $Cl^-$, $ClO^-$ |

| Positive species |
|---|
| $N^+$, $N_2^+$, $N_3^+$, $N_4^+$, $O^+$, $O_2^+$, $O_4^+$, $NO^+$, $NO_2^+$, $N_2O^+$, $N_2O_2^+$, $NO^+(N_2)$, $NO^+(O_2)$, $H_2O^+$, $OH^+$, $H^+(H_2O)_{n=1-7}$, $H^+(H_2O)(OH)$, $H^+(H_2O)(CO_2)$, $H^+(H_2O)_2(CO_2)$, $H^+(H_2O)(N_2)$, $H^+(H_2O)_2(N_2)$, $O_2^+(H_2O)$, $NO^+(H_2O)_{n=1-3}$, $NO^+(CO_2)$, $NO^+(H_2O)(CO_2)$, $NO^+(H_2O)_2(CO_2)$, $NO^+(H_2O)(N_2)$, $NO^+(H_2O)_2(N_2)$ |

| Neutrals |
|---|
| N, $N(^2D)$, $N(^2P)$, O, $O(^1D)$, $O(^1S)$, $O_3$, NO, $NO_2$, $NO_3$, $N_2O$, $N_2O_5$, $HNO_3$, $HNO_2$, HNO, $H_2O_2$, $N_2$, $O_2$, $H_2$, $CO_2$, $N_2(A^3\Sigma_u^+)$, $N_2(B^3\Pi_g)$, $N_2(C^3\Pi_u)$, $N_2(a^1\Pi_g)$, $N_2(a'^1\Sigma_u^-)$, $O_2(a^1\Delta_g)$, $O_2(b^1\Sigma_g)$, $H_2O$, $HO_2$, OH, H, HCl, Cl, ClO |
| New: $CH_4$, $CH_3$, $CH_3O$, $CH_3O_2$, $CH_3OOH$, $CH_2O$, HCO, CO, HOCl, $ClONO_2$, OClO |




**Table 3.** Electric field driven processes additionally included compared to the model of Winkler and Notholt (2014). The reaction rate coefficients in air depending on the reduced electric field strength were calculated with the Boltzmann solver BOLSIG+ (Hagelaar and Pitchford, 2005) using cross section data for $H_2O$ (Itikawa and Mason, 2005), and for $H_2$ (Yoon et al., 2008).

| |
| --- |
| Ionisation |
| $H_2O + e \rightarrow H_2O^+ + 2e$ |
| Electron attachment |
| $H_2O + e \rightarrow OH^- + H\;^\star$ |
| $H_2O + e \rightarrow O^- + H_2$ |
| Dissociation |
| $H_2O + e \rightarrow OH + H + e$ |
| $H_2 + e \rightarrow H + H + e\;^{\star\star}$ |

$^\star$) Sum of $(OH^- + H)$, and $(H^- + OH)$.

$^{\star\star}$) Sum of $(H + H + e)$, $(H^- + H)$, and $(H^+ + H + 2e)$.



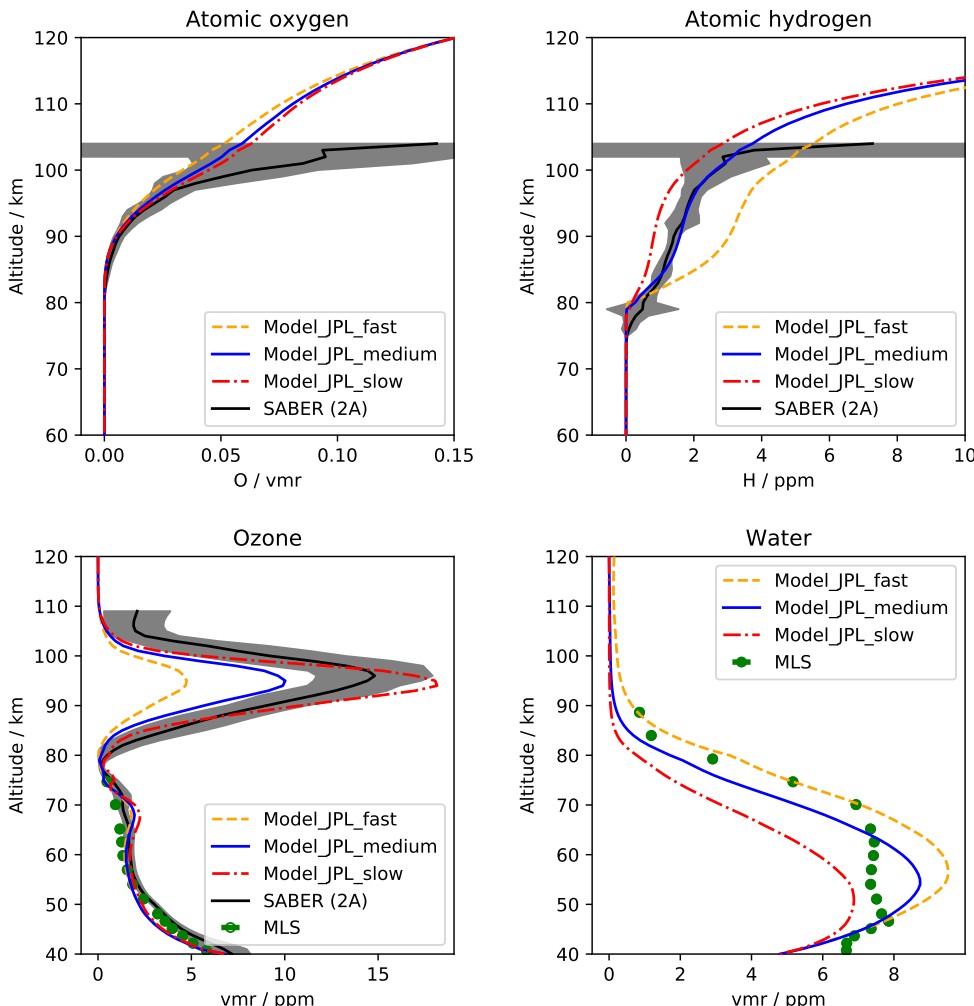

**Figure 1.** Modelled mixing ratio profiles of selected trace gases for the undisturbed atmosphere before the sprite event B in comparison with satellite data. The model results are for 18th of November 2009, 0:15h local time, solar zenith angle 165.7° at 6.7°N, 79°W. For all model runs, JPL rate coefficients were used. Shown are results for slow, medium, and fast vertical transport. Black solid lines show SABER Level 2A (v2) data zonally averaged night-time values for November 18, 2009, and latitudes 0°–13.5°N; the mean solar zenith angles is 160°. The gray areas depict ± one standard deviation of these profiles. Green data points depict MLS (Level 2, v04) zonally averaged night-time values for November 18, 2009, and latitudes 33°S–40°N; the mean solar zenith angle is 142°.



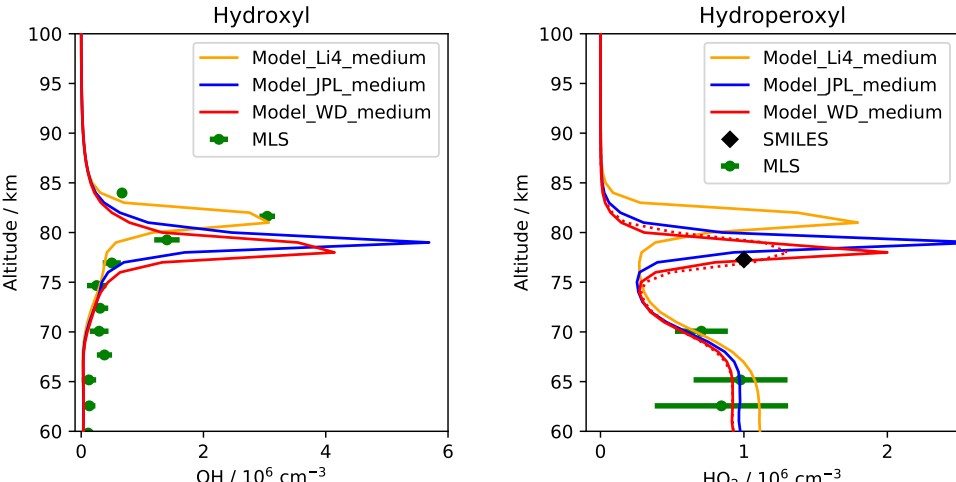

**Figure 2.** Modelled altitude profiles of (left) OH and (right) $HO_2$ number densities before the sprite event B in comparison with satellite data. The solid lines depict model profiles for 18th of November 2009, 0:15h local time, solar zenith angle 165.7° at 6.7°N, 79°W. Shown are results of three model simulations with different rate coefficients for some $HO_x$ reactions (see text for details). For all model runs, medium vertical transport velocities were used. Green data points depict MLS (Level 2, v04) zonally averaged night-time values for November 18, 2009, and latitudes 33°S–40°N; the mean solar zenith angle is 142°; the error bars correspond to one standard deviation. The SMILES $HO_2$ data point corresponds to the atmospheric background value prior to the sprite event measured at 77 km (Yamada et al., 2020). Corresponding to the vertical resolution of SMILES, the dotted red line shows a 3 km running average of the red Model_WD profile.





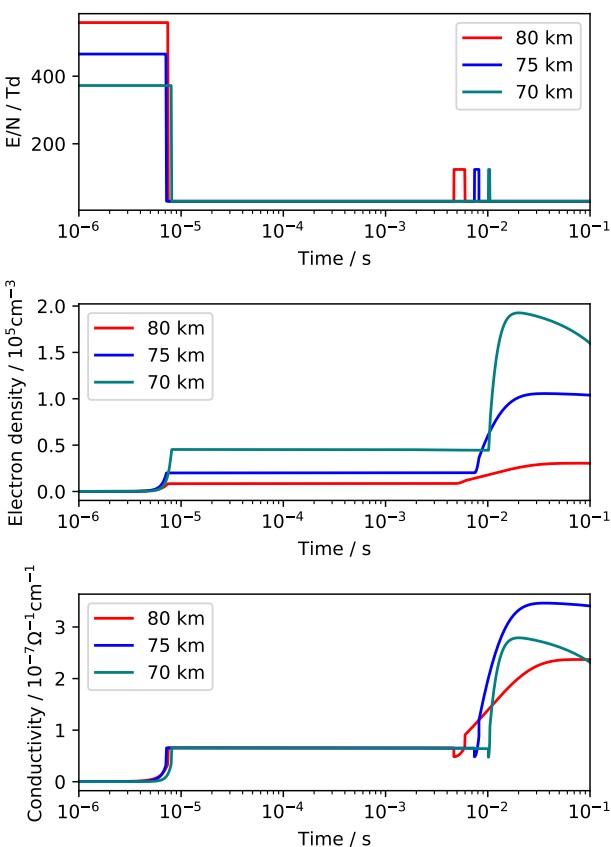

**Figure 3.** Electric parameters of the streamer discharge as a function of time for altitudes 70, 75, and 80 km. Upper plot: Prescribed reduced electric field strength (in units of $\mathrm{Td} = 10^{-17}\mathrm{Vcm^2}$); middle plot: Electron density; lower plot: Electron conductivity.

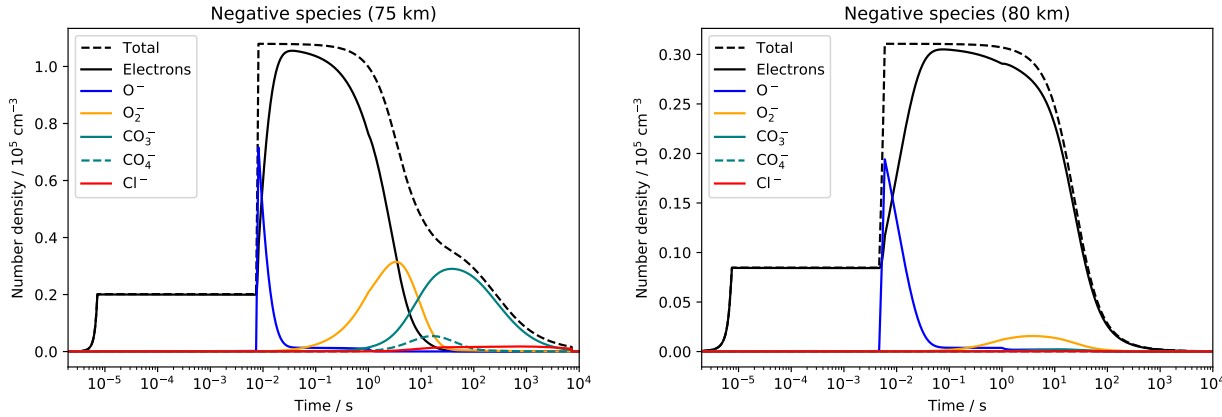

**Figure 4.** Modelled concentrations of the most abundant negative species under the influence of the streamer electric fields as a function of time at (left) 75 km, and (right) 80 km altitude.





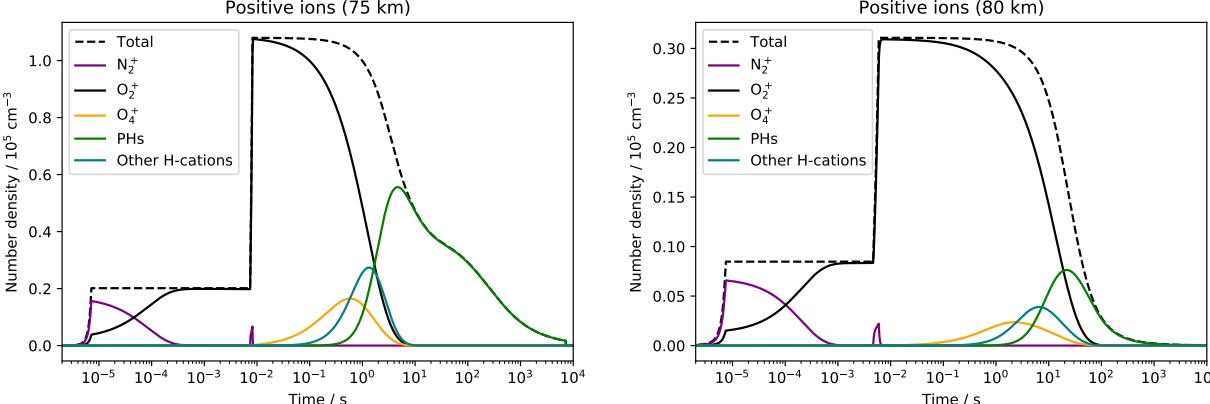

**Figure 5.** Simulated concentrations of the most abundant positive ions under the influence of the streamer electric fields as a function of time at (left) 75 km, and (right) 80 km altitude. PHs denotes the sum of all modelled proton hydrates ($H^+(H_2O)_{n=1...7}$). The teal solid line shows the sum of all hydrogen containing positive ions except for proton hydrates.

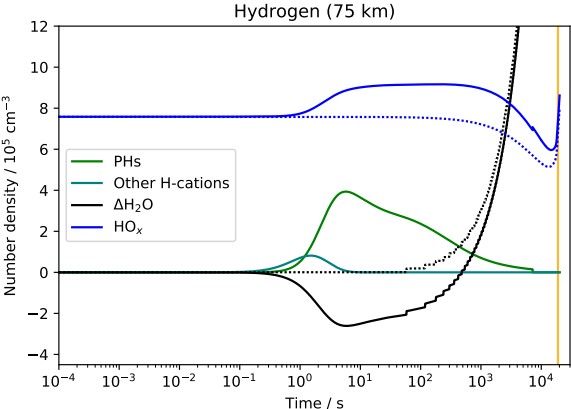

**Figure 6.** Modelled evolution of hydrogen containing species at 75 km altitude. The solid lines show the sprite model run, and the dotted lines show a model run without electric fields applied. PHs denotes proton hydrates. The teal solid line depicts all hydrogen-bearing positive ions except for proton hydrates. Because of its large abundance, for water not the absolute concentrations are shown but the change of the concentration with respect to its initial value. The step like changes of $H_2O$ are due to the transport simulations once every minute. The vertical orange line indicates the time of sunrise.





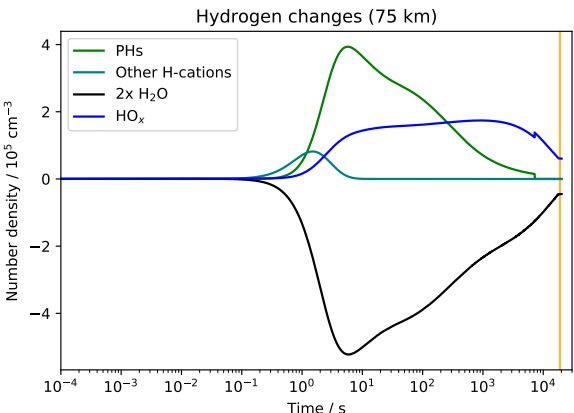

**Figure 7.** Evolution of the modelled amount of hydrogen atoms contained in selected species at 75 km altitude. Shown are differences between the sprite model run and the model run without electric fields applied. PHs denotes all hydrogen atoms in proton hydrates. The teal solid line depicts all hydrogen atoms in positive ions except for proton hydrates. The vertical orange line indicates the time of sunrise.

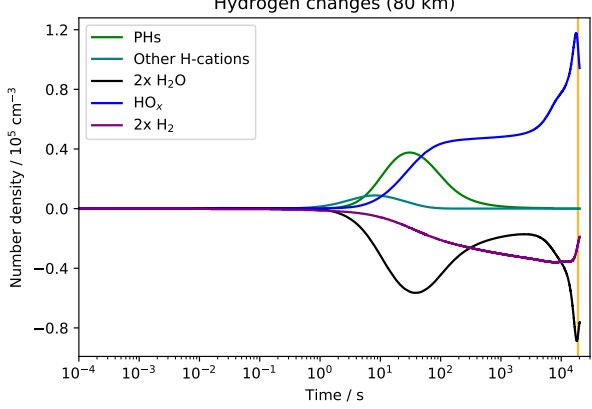

**Figure 8.** Similar as Fig. 7 but here at 80 km, and additionally showing the change of the total hydrogen amount in form of $H_2$ (purple line).





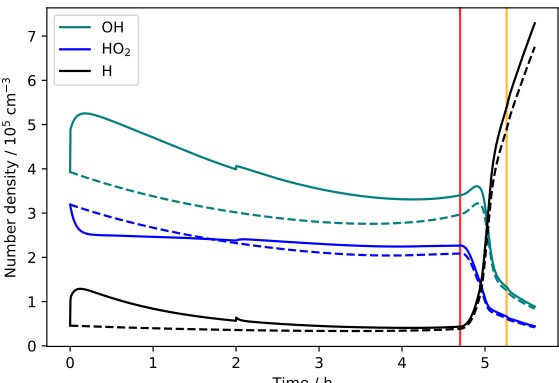

**Figure 9.** Concentrations of H, OH and $HO_2$ at 75 km altitude. The solid lines depict the sprite model simulation, and the dashed lines depict the no-sprite simulation. The vertical orange line indicates the time of sunrise, and the vertical red line indicates the time when the model starts to account for scattered sunlight (at a solar zenith angle of $98°$).

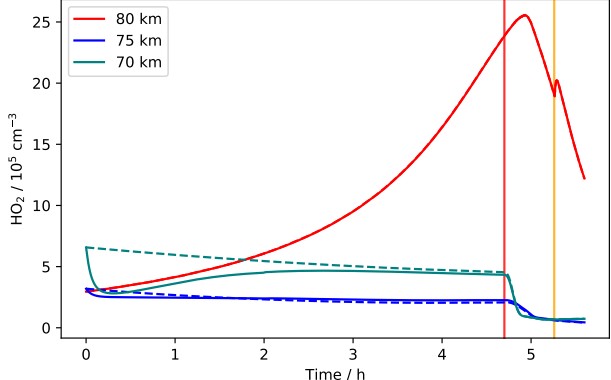

**Figure 10.** Concentrations of $HO_2$ at altitudes 70, 75, and 80 km. The solid lines depict the sprite model simulation, and the dashed lines depict the no-sprite simulation. The vertical orange line indicates the time of sunrise, and the vertical red line indicates the time when the model starts to account for scattered sunlight (at a solar zenith angle of $98°$).

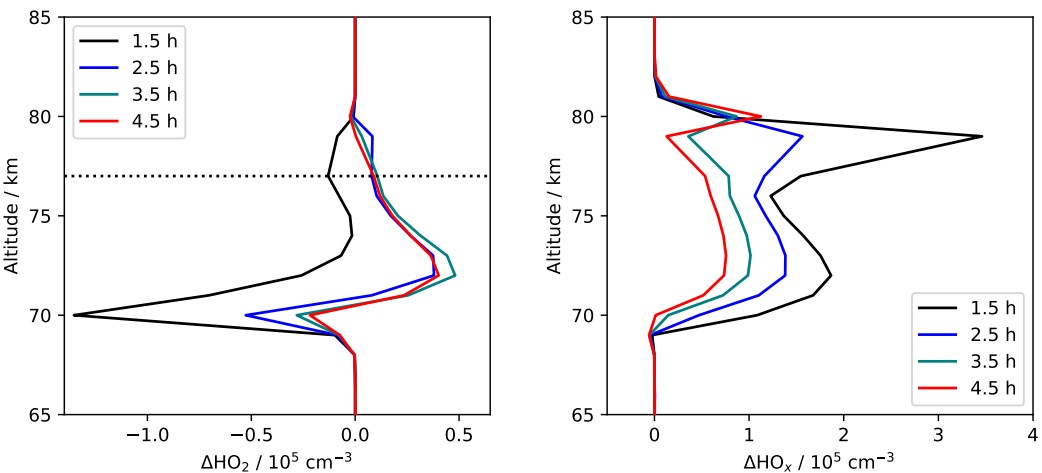

**Figure 11.** Concentration differences between sprite and no-sprite simulation as a function of altitude for different times after the sprite discharge. Left: $\Delta HO_2$; right: $\Delta HO_x$. The dashed lines marks the SMILES measurement tangent height altitude of 77 km.

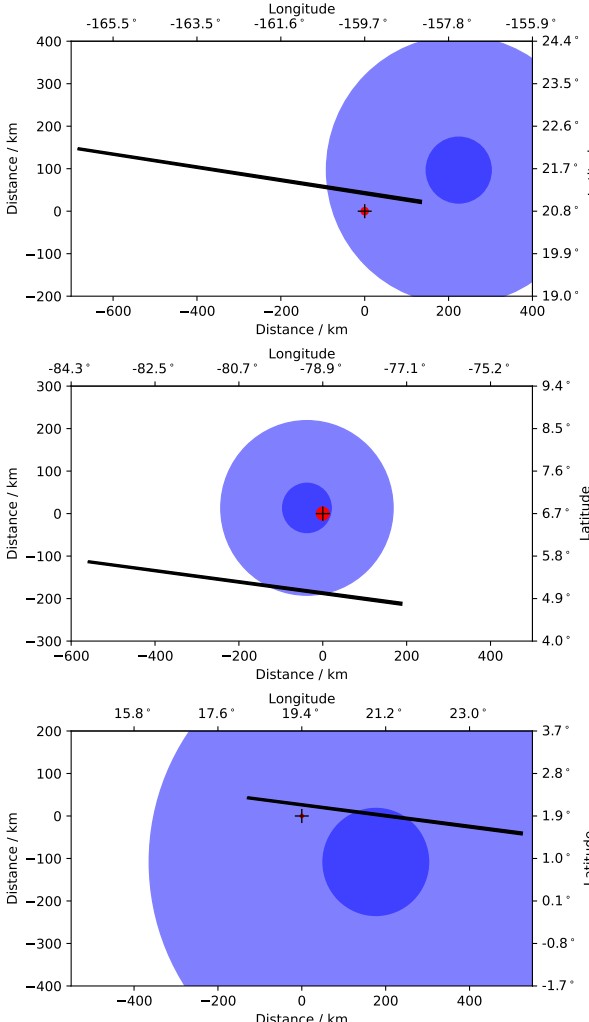

**Figure 12.** Results of the sprite transport and dispersion calculations. From top to bottom: Sprite event A, B, and C (Tab. 1). The axes give latitude and longitude of the scenes as well as the meridional and zonal distances from the initial sprite locations (all three maps display the same area size). The red circles depict the initial sprite cross sections derived from the observed horizontal widths of the sprites. The blue circles indicate the sprite cross sections at the times of the SMILES measurements. The large/small blue circles correspond to the fast/slow diffusion scenario. The black areas show the fields of view of the SMILES measurements.



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
