# Peer review of "Model simulations of chemical effects of sprites in relation with observed HO2 enhancements over sprite producing thunderstorms"

_Atmospheric Chemistry and Physics, 2020_

## Author Comment (AC1)

**Response to the reviewers' comments on acp-2020-1228.**

We thank the reviewers for their valuable comments, questions and suggestions.

Some remarks before we address the reviewers' comments in detail:

As suggested by one of the reviewers, the section on the sprite simulation were split into different sections. As a result, some parts of the original manuscript are now found in different sections.

We modified the way of estimating the total HO2 increase per sprite from the model predictions. We removed the whole part of a single streamer measured by SMILES. For the new estimation we consider the actual volume of the SMILES field of view in the altitude range of the sprite.

This changes some numbers but the general issue remains: The model predicts some HO2 increase, SMILES saw much more. We do not claim that the model simulations explain the observation.

The central part of the summary section now reads:

... the estimated number of sprites needed to explain the observed HO2 enhancements is unrealistically high. The estimated numbers of sprites that occurred near to the SMILES measurement volumes are much lower. The discrepancies increase with increasing measurement tangent height. For the highest tangent height, the model does not predict any HO2 in contrast to the observations. Therefore, in general the model results do not explain the measured HO2 enhancements. At least for the lower measurement tangent heights, the production mechanism of HO2 predicted by the model might contribute to the observed enhancements. It is not clear whether the discrepancies between model predictions and observations are due to incorrect model parameters and assumptions or whether there are chemical processes missing in the plasma chemistry model. ...

**Anonymous Referee #1**

This is an interesting paper showing the efforts made by the authors to model the SMILES measured  $HO_2$  chemical signature associated to sprite streamer chemical activity in the mesosphere (70 - 80 km). What is the final cause of the measured  $HO_2$  increase?. There are no clear conclusions in the paper since available measurements and model results do not completely match. There is not a clear causal link between the enhanced  $HO_2$  observations and the sprite streamer + transport modeling described in this paper. **Response:** Please see the part from the summary section above.

The paper is mostly clear and well written. There are, however, some comments I would like to make.

**Satellite observations**

This section is devoted to briefly explain SMILES measurements from the ISS of enhanced mesospheric HO2 over sprite-producing thunderstorms. Already here the authors indicate that ISUAL detected three thunderstorm systems producing sprites prior to SMILES observations. Authors also highlight that WWLLN indicated strong lightning activity in these 3 thunderstorms systems and that more sprites than those detected by ISUAL could have been occurred. I miss here a thorough discussion about the 3 thunderstorms systems producing the sprites that seem to have triggered HO2 detections by SMILES. In particular, how many positive and negative lightning occurred?. What were their corresponding charge moment change (CMC)? Where (and when) did they occur?. It is known (see Qin et al. GRL 2013) that lightning CMC values can largely determine the type/morphology of sprite (columnar or carrot-like). In particular, CMCs > 500 C km favor carrot sprites, while column sprites (with less streamers) are usually associated to lightning with CMCs lower than ~ 500 C km. I think that an exhaustive analysis of those 3 thunderstorm systems is crucial here because they critically condition the frequency and type of sprites produced. At least one carrot sprite (associated to event A) was reported in Yamada et al. GRL 2020 (see Figure 1d). A sprite halo with downward many propagating streamers looking like the onset of a carrot sprite (event B, see Figure 1e in Yamada et al. GRL 2020) was also detected by ISUAL prior to measurements by SMILES. The image shown in figure 1f of Yamada et al. GRL 2020 also seems to be a carrotlike sprite.

**Response:** The lightning properties such as polarity and CMC were not stored in the WWLLN database which has only about location and time for each detected lightning event during the SMILES observation period. Therefore, it is difficult to estimate the number of sprites. Based on the number of WWLLN lightning strokes in area close to the SMILES measurements we make a rough estimation of the number of sprites at the end of Section 7.

**Sprite chemistry and vertical transport simulations**

**Line 147:** I would replace "*afterglow region*" by streamer glow or streamer trailing glow region. The word "*afterglow*" somehow indicates chemical delayed reactions, but the chemistry in sprite glows are driven by an active electric field (~Ek). In a time-integrated image of a sprite, the image is mostly dominated by the sprite glows (see *Stenbaek-Nielsen and McHarg, JPD-AP 2008*). Streamers only leave relatively faint traces in long exposure images. Thus, if we consider optical emissions as a driver for energy input into the mesosphere, this implies that the main local energy dissipation is in the sprite streamer trailing glows and beads, as studied by *Parra-Rojas et al. JGR-Space Physics* (2015). **Response:** It was changed to streamer glow region

**Line 160-161:** The duration of the glow luminosity (field) can be up to 100 ms (see *Stenbaek-Nielsen and McHarg JPD-AP, 2008*). At 80 km, *Gordillo-Vázquez and Luque GRL 2010* used 8 ms long sprite trailing glows at 80 km. However, the authors use only 1.3 ms, which is a bit too short. *Parra-Rojas et al. JGR-Space Physics (2015)* implemented long (5 ms - 100 ms) sprite trailing glows in a 1D sprite kinetic model. Unfortunately, they did not study the evolution of HO2 species.

**Response:** The major difference between Gordillo-Vázquez and Luque (2010) and our setting seems to be that the first electric field pulses have different duration times and this lead to different electron densities behind the streamer tip (the number densities of seed electron might also be different). We have chosen the duration of the streamer tip pulse so

that the resulting electron density agrees with the values of Luque and Ebert (2010). Because of that the second pulse cannot be much longer than in our simulation without producing too much electrons. We have performed another simulation with increases duration time of the second pulse. This is described in the manuscript now.

What is the impact in the predicted  $HO_2$  concentration (number of molecules) of not considering the streamer glow field (roughly Ek)?.

**Response:** We have also performed a simulation without second field pulse. It is included in the manuscript now.

When discussing large proton hydrates kinetics (page 7) in the D-region, the authors explicitly indicate the key recombination of  $H^+(H_2O)n$  with electrons taking place in the mesosphere but seem to not consider (though mentioned in line 201) proton hydrates recombinations with negative ions. In this regard, the Mitra-Rowe (M-R) scheme consider the kinetics of positive hydrated ions like  $H^+(H_2O)_n$  (see Gordillo-Vázquez et al JGR-Space Physics 2016) applicable to the 70-85 km region. The authors do not seem to consider recombination of  $H^+(H_2O)_n$  with negative ions such as  $CO_3^-$  and  $O_2^-$ . Were these reactions considered?.

**Response:** They are considered. We added some information to that section 5, staring with: **Proton hydrates can undergo recombination reactions with atomic or molecular anions as well. In the model this is accounted for by two-body and three-body recombination processes, for details see the Supplement to Winkler and Notholt (2014).**...

**Line 232:** Suggest to replace: "... the concentration of  $HO_2$  increases." by "... the concentration of  $HO_2$  slightly increases above ambient values." **Response:** Done.

**Line 235-236:** The red dashed line is missing in Fig. 10. **Response:** Actually it is there but it is basically the same as the dashed line.

**Line 255-260:** What types of sprites are reported by *Heavner et al. (2000)*, *Kuo et al. 2008* and *Takahashi et al. 2010*?. The ~10e22 photons per sprite streamer is a reasonble number that agrees with available detailed simulations. The 10e24 photons per sprite could be typical of column-like sprites (with some tens to a few hundreds of streamers).

**Response:** Good to hear that 1e22 is a reasonable number. We didn't find any reference yet. I guess that the cited papers are about column-sprites although this is not stated. The Heavner data goes back to the 1994 sprite campaign, Kuo analysed 155 ISUAL recorded sprites without providing further information, and Takahashi considered 14 "representative sprites".

I agree in that it is unlikely that the measured  $HO_2$  enhancenment is only due to 3 sprites.

Line 284: 38000 (7600) sprite events is completely unrealistic.

**Response:** The numbers have changed a bit due to the different estimation we are doing now. But they are still too large. Please see the rewritten part from the summary section above (or in the manuscript).

As said above, a careful analysis of the the 3 thunderstorms and the produced types (lightning polarities, CMCs, ...) of sprites (column/carrot, producing infrasound?, ...) would be important to advance in the understanding of the underlying reasons leading to HO2 enhancements in the mesosphere.

**Response:** As said above, there is little information but we still tried to exploit them at the end of section 7.

Finally, it would also be interesting if the authors could show a plot of the model predicted ozone ( $O_3$ ) density, whether it is predicted to stay the same, increase or decrease at 75 km, 77 km and 80 km. SMILES did not measure a clear change of  $O_3$  due to sprite chemical activity.

**Response:** The model predicts a short term decrease of ozone followed by a slight increase. This could be shown but we would prefer to keep the paper focused on HOx as the SMILES measurements of O3 are inconclusive.

**Some details:**

What are the branching rations of each channel in: a) H2O + e -> OH + H / OH + H - and b) H2 + e --> H + H + e / H - + H / H + + H + 2e.

**Response:** This depends somewhat on energy. For the streamer tip fields: H2O -> H- + OH dominates (~95%), and H2 -> 2H dominates (~82%) followed by ionisation, attachment is negligible.

Figure 12: Caption: I think the authors mean "black line" instead of "black areas"?.

**Response:** Strictly speaking these are more complicated shapes but "lines" is actually less confusing here, and a good approximation. It was changed. Also, the lines now show the SMILES field view below 81.5 km. (instead of <90km as in the previous manuscript). This appears to be more appropriate when it comes to relate the SMILES measurement volumes to the altitude range of the sprites and the model simulations.

Reaction 12 should be: H + O2 + M - HO2 + M instead of "H + O2 + M - HO2 + M" that would not be well balanced.

**Response:** Yes, that was nonsense. It was corrected.

**Anonymous Referee #2**

Very recently, Yamada et al. (2020) reported first-time observations of mesospheric HO2 enhancements in regions of proven sprite activity. The manuscript by Holger Winkler and colleagues is a timely contribution to give a model interpretation of these observations. It is a very detailed model study of HO2 changes related to sprites and includes a much-needed modeling of the dispersion of the air masses affected by the perturbing events, therefore bridging between sprite-streamer chemistry predictions and air masses actually sounded by the satellite. The observations with Winkler et al.'s interpretation could in principle give a constraint to the several models developed over the past 2 decades on sprite chemistry, a source that is as yet poorly constrained and of interest to the broader atmospheric community. I think the study is well developed and discussed, mostly well written and with high quality figures. There are some improvements that could be applied and I invite the authors to consider the following comments before acceptance for publication in ACP.

**GENERAL COMMENTS**

The main finding of the study is that modelled sprite HO2 cannot explain what sounded by the SMILES instrument, unless an unrealistic number of sprites were contributing. The difference between model and observations is of 3 to 4 orders of magnitudes. Given that typically one expects a few tens of sprites over a thunderstorm (in a relatively compact volume since it is sounded by one SMILES measurement), a few orders of magnitude difference persists. I miss a thorough discussion of what factors are at play in the model that limit the HO2 production. Several factors are then cited as possible shortages, although there was no quantitative analysis of what parts of the study could lead to order of magnitude increases. I think such a detailed study could really give guidelines on where the discrepancies are to be found. **Response:** We were not able to come up with "the solution" but we added some words to the

discussion section 8.

In Yamada et al., Fig 2, there seems to be a tiny decrease in ozone consistent among the three cases. Even though very limited, would a decrease be consistent with model predictions? Is this the only further species detected by SMILES? It would be of great help to look also at other species, which may help to better relate observations and model. predictions.

**Response:** For ozone it is difficult to distinguish between natural fluctuations along the LOS and a depletion at the event area due to its rich abundance. We do not consider these single measurements to be significant. (The model predicts a short term decrease followed by an increase but we would prefer to keep the paper focused on HOx.) Regarding the other species we added this to Section 2: HO2 is the only active radical for which an effect was observed. SMILES spectra of H2O2 and HNO3 have been analysed but there are no perturbations around the events due to very weak line intensities.

The observational uncertainties are only shortly introduced in the table. I think there is a need to further explore these uncertainties to help reconciling observations and model predictions. How are the observing geometries affecting Yamada et al. estimates? Could there be a contamination of the HO2 spectral features? How is the sprite HO2 production further diluted in the large volumes sounded by the instrument along its lines of sight?

**Response:** There are orders of magnitude differences between observations and model predictions. We don't think that these differences can (to a large extent) be attributed to measurement errors. Of course, we agree that a better estimation of measurement

uncertainties would be desirable. However, there is little additional information we can provide compared to the paper by Yamada et al. We added this to the manuscript: As shown in Figure S2(d-f) in the supporting information of Yamada et al. (2020), the retrieved total HO2 enhancements are basically independent on the assumed volumes in which HO2 is increased. The authors evaluated the impact of a possible contamination of the spectral HO2 features on the retrieved HO2 enhancement to be of the order of 10–20%.

Furthermore, the transport study shows that only fractions of the air masses affected by the sprites are sounded, but no quantitative consideration is made of its further dilution effect. Are these expanded/transported airmasses consistent also with a multiple-sprite scenario? The apparent dilution along the line of sight should be considered also in this case.

**Response:** We have changed these estimations (second half of Section 7) and are now taking into account the volume expansion of the sprite air masses and the volumes of the SMILES field of views.

**DETAILS**

Title: I would find the title more attractive if it represented better the focus on HO2 **Response:** We agree, and changed it to: **Model simulations of chemical effects of sprites in relation with observed HO2 enhancements over sprite producing thunderstorms**

L41: "These are the first direct observations of chemical sprite effects". I would be more careful with such statements. Yamada et al. were the first observations of HO2 enhancements in regions of proven sprite activity, not direct measurements of chemical changes through a sprite. The lack of consistency between model and observations seem to further require this caution.

**Response:** We completely agree on that. That sentence was replaced by: **The aim of this paper is to compare these observations to model simulations of chemical sprite effects**

L44: A few words of comments would be helpful on the decrease predicted for HO2 by Hiraki et al. 2008. Isn't this relevant to Yamada et al. observations? Yamada et al. reported observations up to 80 km altitude so some cases would see a reduction of HO2 whereas the other cases an increase?

**Response:** This was an error in the manuscript. It is the other way around. Hiraki et al. predicted an *increase* of HO2 at 80 km, and a *decrease* at lower altitudes. This has been corrected.

L45: Yamada et al. 2020 already presented model predictions but these are not mentioned here in the introduction. Why? It should be clarified whether the model and simulations presented in this manuscript are different (and how) from those presented in Yamada et al. **Response:** We added / modified in the introduction: **Yamada et al. (2020) have presented preliminary model results of an electric field pulse at 75km which indicate an increase of HO2. In the present paper, we show results of an improved sprite chemistry and transport model ...**

L60 and around. The observational results are affected by uncertainties, which are only reported in table 1 and not presented in the paragraph. Because of the discrepancies found between model and observations, I would find it useful to anticipate here a detailed description of all possible sources of these discrepancies. For example, limb sounding

measurement is affected by spread of information along the line of sight. How large is this spread? How are the averaging kernels? 3-400 km as for other instruments? What is the pointing error? How accurate is the geolocation? It is mentioned that Yamada et al. estimated advection of a few 100 km. In what direction?

**Response:** In Yamada et al, the advection is in longitudinal direction due to the zonal winds. As we report on improved transport simulations in this paper, we did not add any further information to the Yamada calculations here. Because the averaging kernels represents the sensitivity of the retrieved state with respect to the true state, Yamada et al., only tested the influence of the change of the spread of the information along the line-of-sight. We added: As shown in Figure S2(d-f) in the supporting information of Yamada et al. (2020), the retrieved total HO2 enhancements are basically independent on the assumed volumes in which HO2 is increased. The authors evaluated the impact of a possible contamination of the spectral HO2 features on the retrieved HO2 enhancement to be of the order of 10–20%.

L69 it's - -> its **Done.**

L94 data from SABER are used as climatological background. Are there no other measurements directly from SMILES? Please add a comment. **Response:** No useful SMILES profiles were available to us.

L120 the impact of changing vertical transport speed in the model\_JPL estimates is very large. H2O at 80 km altitude (i.e. one of the case studies) changes from 1.5 to 4 ppmv. Large differences are found as stated/shown also in ozone and atomic hydrogen. It may be difficult for the reader to understand here and in the following whether these large discrepancies have an impact or not. I assume that water abundance is so large that these starting differences have no impact, so I would anticipate it here.

**Response:** We did the simulations with different transport velocities to check if this affects the sprite model results. The short answer is: No, basically not. A bit longer one is now given in section 8: The model relies on prescribed vertical transport parameters. A variation of the transport velocities has significant impact on the altitude profiles of long lived species including H2O (see Fig. 1) which potentially can affect the HO2 formation in a sprite. (...) We have repeated the (...) simulation with faster and slower vertical transport. The effect on the sprite induced HOx production and HO2 enhancements is very small. At all altitudes, the abundance of H2O is much larger than the produced amount of HOx. Water is not a limiting factor for the formation of HO2.

L140 MLS data points were averaged over a very large region. It seems therefore appropriate to give an estimate of the variability of these measurements. Since this works attempts to describe conditions found in the three case studies, it is essential to understand the range of background conditions that could be reasonably found and how these impact on the results: therefore, the scatter should be considered, both due to measurement errors and actual natural variability.

**Response:** The purpose of these comparisons with measurement data was to show that the model does an OK job of reproducing the main properties of some key species' profiles, and to adjust the transport parameters. For SABER a smaller latitudinal range worked than for MLS. We have tested all slow/medium/fast transport cases with all three sets of rate coefficients. The sprite model results are basically not affected. We mention this in Section 8

now, see your previous question and the part starting with: In order to test for the effects of changed rate coefficients, we have also performed sprite simulations using Model\_JPL and Model\_Li4 ...

L141 Section 5. This section is very rich and the full description with no breaks become very difficult to follow. I recommend introducing subsections or an alternative approach to split the flow into a few blocks to help the reader to quickly understand the main points.

**Response:** Good idea. We rearranged it. Section 5 is now only on the sprite streamer simulations.

L220 and following. How this compares to the findings by Hiraki 2008? Were there similar mechanisms linked to the changes at 80 and 70 km altitude?

**Response:** As already stated earlier, we had to correct our statement concerning Hiraki et al. In the introduction. In Section 8 we included: (...) The increase of HO2 at 80 km predicted by Hiraki et al. (2008) is not in contrast to our model results. Also our simulations show such an increase of HO2 at high altitudes but this is just an effect of the continues formation of HO2 in the upper mesosphere during night. It also takes place in the model simulation of the undisturbed atmosphere without sprite discharge.

L240 Since there is such a stringent constrain on the timing of the SMILES measurement and previous sprite activity, an analysis of lightning activity of the three thunderstorms would be very helpful. Can we reasonably expect sprites in the few hours prior to the SMILES passage? This is mentioned in

**Response:** We roughly estimate an expectation value of the number of sprites at the end of section 7.

L274-281 but only qualitatively. Given the relevance of this point a quantitative estimate should be considered.

**Response:** We removed that here. See you question on the issue of accumulation in the summary section below.

L241-245 SMILES cases A and C had tangent points at 75 and 80 km altitude. Why mentioning only the 77 km one? The discussion continues focusing on case B. It would be useful to clarify this and add a comment on the other cases studies.

**Response:** We haven't changed this here but now consider the three cases in the second half of Section 7.

L245 I would split section in subsections for example here. **Response:** We did so.

L248 Is the 850 m diameter consisting of a volume completely filled by an individual streamer channel or simply be a volume with a variety of branches of different scales? Would this change the estimates that follow? I would specify this is the text. **Response:** We removed that one-streamer thing completely.

L252 This is a clever approach. How robust and variable are these estimates? Are photons from the internal parts of the sprite expected to escape undisturbed or should one expect an onion-like shielding effect? Can this increase the amount of excess HO2 molecules? Is this approach better than that used by Arnone et al. 2014 that was cited? They used the current

moment, shouldn't the two approaches lead to consistent estimates? This is a key point in this study and I feel it should be better explored in its limitations and giving a possible range of the adopted estimates.

**Response:** Good questions. Difficult to answer without radiatitive transfer simulations and without knowing about the absorption cross sections of the streamers (probably high?). My feeling is however, that due to the rather low filling fraction of the streamer, a large portion of photons may escape. We just added **neglecting absorption of photons inside the sprite volume** to that sentence. Considering the current moment is an interesting approach as well but I'd say that the number of emitted photons is a good closer related to the chemical processes we are interested in.

L263 There is no mention of the direction along the line of sight. The signal is being integrating over likely a few hundred km (please give robust estimates for this), so that a further important dilution of the predicted sprite HO2 enhancement occurs (likely of the order of 30 km / 300 km, which is a factor 1/10). This decreases the amount of enhancement that SMILES would have seen due to a single sprite. I think this is an important point that was missed and should be quantified.

**Response:** We modified that estimation. Now we use the volume of the SMILES field of view in the sprite altitude region. We think this is better. It also causes differences between the different tangent heights. Please read the second half of Section 7.

L266-273 there is little effort in estimating how and by how much a larger HO2 enhancement could be obtained. I think the three points that were identified "missing chemical processes considered by the streamer model, inaccurate electric field parameters or reaction rate coefficients" should be further investigated giving quantitative estimates. For example, the very interesting approach of multiple models shows that the different rate coefficients considered have no significant impact (only a change in the first couple of hours). **Response:** This part was rewritten, please see Section 8.

L277-280 Here the 3 cases are recalled, although no mention is made of case C at 80 km tangent altitude. In Yamada et al. 2020, also case C shows clear enhancements of HO2, how would this be possible given the negligible predicted HO2 production?

**Response:** We simply don't know. In the summary part we state: ... The discrepancies increase with increasing measurement tangent height. For the highest tangent height, the model does not predict any HO2 in contrast to the observations.

L282 The authors discuss the possibility that a large number of sprites contributed to the observed HO2 enhancement. This point certainly deserves a discussion but given the 3 or 4 orders magnitude difference between the modelled sprite HO2 production for 1 sprite and that observed by SMILES, "large" is rather unrealistically large. I suggest reviewing the text to make clear since the beginning that one could expect a few tens of sprites per thunderstorms (up to a few hundred in extraordinary cases) and so 3 or 4 orders of magnitude differences cannot be reconciled.

**Response:** (This was rewritten, the numbers changed a bit). In the Summary we state: **Due to the modelled long-lasting increase of HO2 after a sprite streamer discharge, an accumulation of HO2 produced by several sprites appears possible. However, the**  estimated number of sprites needed to explain the observed HO2 enhancements is unrealistically high. The estimated numbers of sprites that occurred near to the SMILES measurement volumes are much lower. It continues with the response to your last question.

L291 I would discuss this part in terms of the air masses interested by the sprite event rather than introducing the expansion of the sprite body since the sprite lasts a few milliseconds. **Response:** This might indeed be a more accurate term. We replaced sprite body/volume by sprite air masses almost everywhere.

Fig 7 and 8. Could you add a thin line at zero? **Done.**

Fig 11. Could you please add a thin vertical zero line? Why is only the SMILES 77 km altitude tangent point plotted in the graph? The three case studies are at 75, 77 and 80 (cases A, B and C respectively). I think having all the three lines would be more appropriate. **Done.**

Fig 12: It would be helpful to report the time difference between the sprite event and SMILES measurement directly in the figure. Also, the figure could be completed adding a contour map of a snapshot of horizontal winds.

**Response:** We have included the time difference in the three plots. The wind fields would look rather boring as there is not much variation over the shown domains. As a result, the sprite centers are moved pretty much on straight lines. Instead of plotting wind data we indicate the displacement and the distance in the plots. We made an additional change: The SMILES field of view now shows the part below 81.5 km. (instead of <90km as in the previous manuscript). This appears to be more appropriate when it comes to relate the SMILES measurement volumes to the altitude range of the sprites and the model simulations.

---

## Author Response (AR2)

**Response to the editor's comment on acp-2020-1228.**

As suggested we have made a change to the abstract text:

Old:
However, the estimated number of sprites needed to explain the observed $HO_2$ enhancements is unrealistically large. The estimated number of sprites that occurred near to the SMILES measurement volumes is much smaller.

New:
However, the number of sprites needed to explain the observed $HO_2$ enhancements is unrealistically large.